# A Closer Look at Dexamethasone and the SARS-CoV-2-Induced Cytokine Storm: In Silico Insights of the First Life-Saving COVID-19 Drug

**DOI:** 10.3390/antibiotics10121507

**Published:** 2021-12-08

**Authors:** Paul Morgan, Shareen J. Arnold, Nai-Wan Hsiao, Chih-Wen Shu

**Affiliations:** 1Faculty of Science and Technology, University of Belize, Belmopan 501, Belize; paul.marcel.morgan@gmail.com (P.M.); sarnold@ub.edu.bz (S.J.A.); 2Department of Biology, National Changhua University of Education, Changhua 50007, Taiwan; nady@cc.ncue.edu.tw; 3Institute of BioPharmaceutical Sciences, National Sun Yat-Sen University, Kaohsiung 80424, Taiwan; 4Department of Biomedical Science and Environmental Biology, PhD Program in Life Science, College of Life Science, Kaohsiung Medical University, Kaohsiung 80708, Taiwan

**Keywords:** dexamethasone, SARS-CoV-2 (COVID-19), cytokine storm, acute respiratory distress syndrome (ARDS), molecular dynamic (MD), inflammatory markers, receptors

## Abstract

The term cytokine storm refers to an uncontrolled overproduction of soluble inflammatory markers known as cytokines and chemokines. Autoimmune destruction of the lungs triggered by the release of these inflammatory markers often induces acute respiratory distress syndrome (ARDS). ARDS is an emergency condition with a high mortality rate in COVID-19 patients. Dexamethasone is the first repurposed corticosteroid with life-saving efficacy in patients with severe SARS-CoV-2 infection. Dexamethasone has traditionally been known to suppress the production of inflammatory markers at the transcriptional level, but its role as a direct therapeutic to neutralize cytokines, chemokines, their receptors, and functionally critical SARS-CoV-2 proteins has not yet been explored. Herein, we demonstrated that dexamethasone binds with high affinity to interlukin-1 (IL-1), IL-6, IL-8, IL-12, IL-21, INF2, TGFβ-1, INF-γ, CXCL8, some of the receptors, IL-1R, IL-21R, IFNGR, INFAR, IL-6αR-gp130, ST2 and the SARS-CoV-2 protein NSP macro X, and 3CLpro, forming stable drug–protein complexes. Our work implied that dexamethasone has the potential to directly neutralize inflammatory markers, further supporting its life-saving potential in patients with severe manifestations of COVID-19.

## 1. Introduction

Severe acute respiratory syndrome coronavirus 2 (SARS-CoV-2) has infected a quarter of a billion persons worldwide, resulting in over 5 million deaths as of 1 October 2021. Although several vaccines are now available worldwide, the perpetual threat from the emergence of new variants has resulted in the COVID-19 pandemic still being one of the greatest public health crisis in modern history. Though two years have elapsed since the emergence of COVID-19, there are still many perplexing questions concerning the pathogenesis and clinical course of the disease [1]. For instance, why are some patients asymptomatic, while others experience more severe forms of the disease, including acute respiratory distress syndrome (ARDS) [1]? The defining pathologic feature of COVID-19 is described as an acute pneumatic process with extensive radiologic opacity, inflammatory infiltrates, diffuse alveolar damage, and microvascular thrombosis [1]. Autoimmune destruction of the lungs triggered by the release of proinflammatory cytokines is emerging as a significant contributor to the morbidity and mortality in patients infected with SARS-CoV-2 [2,3].

An uncontrolled overproduction of soluble inflammatory markers, termed a “cytokine storm”, is largely responsible for inducing ARDS [1]. ARDS is an emergency condition with an estimated mortality of approximately 40%. ARDS patients primarily present with bilateral lung infiltrates and severe hypoxemia [1]. The cytokine storm progresses quickly and the immune system starts attacking the body, resulting in multiple organ failure and eventual death [1]. Several proinflammatory cytokines (IFN-α, IFN-γ, IL-1β, IL-6, IL-12, IL-18, IL-33, TNF-α, and TGFβ) and chemokines (CXCL10, CXCL8, CXCL9, CCL2, CCL3, and CCL5) have been implicated in the aberrant systemic inflammatory response [1]. The term “cytokine storm” originated from the observation that COVID-19 patients requiring intensive care unit (ICU) admission presented with much higher concentrations of TNF-α, CXCL10, and CCL2 in comparison to those with a less severe manifestation of the disease in which ICU admission was not necessary [1,3].

Corticosteroids were discovered in the 1940s and became one of the most effective and widely used treatments for autoimmune disorders and various inflammatory conditions [4]. Corticosteroids have also been routinely used to treat severe respiratory illnesses such as uncontrolled asthma and acute exacerbations of chronic obstructive pulmonary disease (COPD) [4]. Dexamethasone is a synthetic corticosteroid with broad-spectrum immunosuppressive potential that is a synthetic mimetic of the naturally occurring hormone cortisol [2,3]. Corticosteroids are known to possess powerful anti-inflammatory properties, as they are capable of reducing the gene transcription of cytokines and chemokines [3]. Dexamethasone is the very first repurposed corticosteroid to demonstrate life-saving efficacy in patients with COVID-19 [5]. In a large-scale UK-based study known as the randomized controlled trial (RCT) of COVID-19 treatments, 2104 patients infected with SARS-CoV-2 were administered 6 mg of dexamethasone daily for up to 10 days [2,3]. This group was then compared to 4321 patients who received the usual standard of care. The incidence of mortality in the dexamethasone group was reduced by 35% as compared those in the usual care group among mechanically ventilated patients [2]. Moreover, dexamethasone is an inexpensive and readily accessible drug that can be acquired without a prescription in many countries.

Dexamethasone’s mechanism of action is known to upregulate the transcription of anti-inflammatory genes (transactivation), while it also downregulates (transrepression) the transcription of inflammatory genes responsible for producing many proinflammatory cytokines and chemokines, various critical enzymes, and cell adhesion molecules as part of a robust immune response [4]. However, very few studies have probed into the molecular mechanisms of dexamethasone as an efficacious SARS-CoV-2 drug. Herein, we explore the repurposed potential of dexamethasone as a therapeutic drug against several inflammatory markers, their receptors, and a few key SARS-CoV-2 proteins.

## 2. Materials and Methods

### 2.1. Cytokines, Chemokines, Receptors, SARS-CoV-2 Proteins, PDB Files

*Cytokines*: IFNα1 (PDB: 3UX9), IL-1B (PDB: 5MVZ), IL-12 (PDB: 1F45), IL-18 (PDB: 3WO3), IL-33 (PDB: 2KLL), INFa2 (PDB: 4Z5R), INF-γ (PDB: 1FYH), TGFβ1 (PDB: 5vqp), TGFβ2 (PDB: 4KXZ), TGFβ3 (PDB: 1TGK),TNF-α (PDB: 1TNF), IL-1a (PDB: 1IRA), IL-6 (PDB: 1ALU), IL-8 (5D14), IL-21 (PDB: 2OQP). *Chemokines*: CCL1 (PDB: 1DOK), CCL2 (PDB: 2LIE), CCL3 (PDB: 3FPU), CCL5 (PDB: 6AEZ) CCL (PDB: 1RTN), CXCL8 (PDB: 1ILQ), CXCL 10 (PDB: 1O80). *Receptors:* IL-21R (3TGX), IL-1R (1IRA), scFv GC1009 (4KV5), IFNGR (6E3L), INFAR (2HYM), ST2-IL-1RAcP(5V14), ST2(4KC3), MGSA (IL-8) (1ROD), CXCR1 (CXCL8) (6XMN), IL-6:IL-6*αR*-gp130 (1P9M). *SARS-CoV-2 Proteins*: main protease, 3CLpro, (PDB: 6M03), Nsp3 macro X domain (PDB: 6WEY), nucleocapsid protein (PDB: 6M3M), ORF7A encoded accessory protein (PDB: 6W37), receptor binding domain (RBD) (PDB: 6VW1), RNA-dependent RNA polymerase (PDB: 6M71), IL-33:ST2-IL-1RAcP (5VI4), INF-γ:IFNGR (6E3L), IL-21:IL21R (3TGX), INFA2:INFAR (2HYM), IL-6:IL-6A*R*/gp130 (1P9M), IL-1RA:IL-1R (1IRA). The cytokines, chemokines, and SARS-CoV-2 proteins were retrieved from the RCSB Protein Data Bank on 1 May 2021, while the receptors were retrieved on 10 October 2021 [6].

### 2.2. Docking of Dexamethasone

A total of 15 cytokines, 6 chemokines, 11 receptors, and 6 critical SARS-CoV-2 proteins were prepared for docking in BioSolveIT SeeSAR [7]. The biding site mode was used to identify binding pockets and active sites [8]. Dexamethasone was loaded as an SDF file and prepared in the docking mode [9]. A total of 100 poses were generated for each PDB file. The FlexX docking functionality in SeeSAR used an incremental construction algorithm to place and fit dexamethasone into the binding pocket [10]. The HYdrogen bond and DEsolvation (HYDE) algorithm calculated realistic free energies based on desolation and physical interactions of dexamethasone in the binding pocket [10]. The HYDE scoring function was employed to estimate and rank binding affinities for all poses [7]. HYDE’s ability to visualize the ΔG contributions of the individual atom of a ligand inside the binding pocket allowed for reliable discernment and stratification of the most promising poses [9,10,11]. The top 3 poses were for each PDB were selected for further structure–activity relationship optimization [12].

### 2.3. MD Simulations of Dexamethasone Complexes

A total of 15 cytokines, 6 chemokines, and 6 critical SARS-CoV-2 protein–dexamethasone complexes were prepared for MD simulations by cleaning the structures, fixing the side chains, and minimizing the energies. The system was solvated in transferable intermolecular potential with 3 points (TI3P) after performing CMIP titration, the total system charge was equalized by adding ions and water molecules. Initial energy minimization was performed using the steepest decent algorithm until the system converged at Fmax 500 kJ/(mol∙nm). Ions and water molecules were allowed to equilibrate around the protein complexes in two phases. The first phase of equilibration was at a constant number of particles, volume, and temperature (NVT). The second phase was at a constant number of particles, pressure, and temperature (NPT). The system was allowed to equilibrate at a reference temperature of 300 K, or reference pressure of 1 bar for 2.5 ps at a time step of 2 fs. The production simulations were performed for 50 nanoseconds with 2 fs time intervals at 310 K (37 °C), which was representative of the average human body temperature. The root-mean-square deviation (RMSD) was calculated for all cytokine, chemokine, and SARS-CoV-2 protein complexes. Average structures were generated as Protein Data Bank (PDB) files and Audio Video Interleave (AVI) files for simulation playback. Data analysis and correlation plots were generated using GraphPad Prism 5 software (GraphPad Software Inc., La Jolla, CA, USA). 

## 3. Results and Discussion

### 3.1. Implications of Dexamethasone on Cytokine and Chemokine Suppression

While it is well documented that dexamethasone exerts its action primarily at the level of transcription, the binding effects of dexamethasone on cytokines, chemokines, and SARS-CoV-2 viral proteins remains unclear. To evaluate the biding affinity of dexamethasone on the inflammatory proteins involved in COVID-19, including proinflammatory cytokines, chemokines, and their receptors, docking for the structurally binding of this drug on these proteins was employed (Figure 1 and Appendix A). Of the 15 cytokines, 5 chemokines, and 11 receptors that were screened, dexamethasone exhibited the greatest binding affinity for ST2 (Il-33 receptor), IFNGR, IL-21R, IL-1R, INFAR, IL-6*αR*, IL-1RA, IL-12, IL-1, TGFβ1, CCL5, INF*α*2, IL-33, IL-8, CXCL8, and IL-6, whereas the binding potential of drug on the other proteins was not observed (Figure 2 and Table 1). Our work also suggested its role as an inhibitor with high binding affinity, blocking multiple cytokines and chemokines from accessing their target inflammatory receptors.

IL-21 is closely related to IL-2 and IL-15, and is involved in the proliferation and maturation of B cells, T cells, and natural killer cells (NK) [13]. IL-12 induces the production of interferon-γ (IFN-γ), differentiation of T helper 1 (T_H_1) cells, and regulates T-cell and natural-killer-cell responses [14]. IL-12 is a key link between adaptive immunity and innate resistance [14]. Dendritic cells are the primary producers of IL-12 in response to pathogens, such as SARS-CoV-2, during infections [14]. Currently, there is very little evidence on how COVID-19 affects patients treated with cytokine inhibitors [15]. Targeting cytokines that are involved in antiviral responses can theoretically suppress SARS-CoV-2 clearance. However, the effect of suppressing uncontrolled cytokine production with dexamethasone in ARDS may be extremely beneficial in patients with severe manifestations of COVID-19.

IL-1 is a potent mediator of fever, pain, and inflammation against infection [16]. Monocytes, tissue macrophages, and dendritic cells are the primary producers of IL-1 [16]. IL-1 is considered to be an endogenous pyrogen, as it regulates the activity of the hypothalamus, which leads to an increase in body temperature [16]. Expression of adhesion factors is also regulated by IL-1, which allows for the migration of immune competent cells such as lymphocytes and phagocytes to the site of infection [16]. The IL-1/IL-6 pathway is highly upregulated in patients with severe COVID-19 [17]. IL-6 is largely implicated in its response to acute inflammation. The primary producers of IL-6 are T lymphocytes, B lymphocytes, dendritic cells, macrophages, mast cell monocytes, and many other nonlymphocytic cells [18]. Studies have demonstrated that serum levels of IL-6 have been found to be significantly elevated in COVID-19 patients, and the circulating levels correlated positively with the severity of the disease [1]. Thus, binding IL-1 and IL-6 with dexamethasone might potentially prevent respiratory failure in COVID-19 patients [17].

The role of IL-8 is to attract and activate neutrophils in regions of inflammation [19]. Endothelial cells, epithelial cells, macrophages, and even airway smooth muscle cells produce this inflammatory cytokine [19]. Emerging studies suggests that elevated levels of IL-8 in COVID-19 patients lead to hyperreactivity of the airway, resulting in an airway constriction and ultimately ARDS [19]. Thus, the potential benefit of dexamethasone as an IL-8 suppressing agent may be very beneficial to patients with severe SARS-CoV-2 infections. 

In 2008, Mogensen et al. postulated that the array of dexamethasone targets with regards to cytokine production are, at best, only partially characterized [20]. They further showed using an enzyme-linked immunoassay (ELISA) that dexamethasone significantly suppressed IL-6 and IL-8 mRNA production [20]. The strongest experimental evidence yet came from an ELISA-based study by Monick et al., who demonstrated that dexamethasone inhibited IL-1 and TNF-α release of IL-6 and IL-8 from cells in a dose-dependent manner [21]. They proposed that in this instance, dexamethasone inhibition was not, by disrupting the signal transduction pathway, responsible for increasing or decreasing expression of IL-1 or TNF-α receptors [21]. Instead, they postulated that “it is possible that dexamethasone decreases cytokine production by simply freezing the receptor ligand on the cell surface” [21].

IL-33 is a major player in type-2 innate immunity. During cell necrosis or cell damage, many immune competent cells, including macrophages, mast cells, fibroblasts, and dendritic cells, produce IL-33 [22]. Its role is to alter the immune system after epithelial or endothelial cell damage during infection and trauma [22]. Not surprisingly, COVID-19 patients with severe manifestations of the disease often suffer from diffuse alveolar damage [23]. Once again, the inhibitory potential of dexamethasone against IL-33 might reduce SARS-CoV-2-induced lung damage in COVID-19 patients.

CCL5 is a potent chemoattractant for B cells, monocytes, T cells, eosinophils, dendritic cells, and natural killer cells [24]. Its role as a chemokine is to recruit leukocytes into inflammatory tissues [25]. CCL5 also stimulates basophils to release histamine and eosinophils to secrete cationic proteins [25]. Histamine triggers an aberrant immune response, leading to cytokine storms and ultimately multiple organ failure [26]. Our work demonstrated that dexamethasone binds with high affinity to CCL5 and forms a stable drug–ligand complex. The binding of uncontrolled CCL5 production in COVID-19 patients might potentially reduce the incidence of severe lung damage due to inflammatory responses, leading to lower morbidity and mortality.

In 2019, Kadowaki et al. demonstrated that pH-stimulated CXCL8 production in human airway smooth muscle cells (ASMCs) was inhibited by dexamethasone in a dose-dependent manner [27]. Zhang et al. (2021) devised a Western blot strategy to show that levels of intracellular IL-6 modestly increased when cells were treated with exogenous recombinant IL-6 [28]. However, treatment with recombinant IL-6 and dexamethasone resulted in a 70% decrease in immunoreactive IL-6 proteins when compared to the untreated cells [28].

IFNα2 is a cytokine produced by cells infected by viral agents [29]. It also acts locally and systemically on other cells to prevent them from a viral infection [29]. Although IFNα2 suppresses proinflammatory pathways and cytokine production, improper regulation can polarize the immune system, leading to severe adverse effects [30]. Dexamethasone’s inhibitory potential against IFNα2 may be very beneficial in reducing the severity of COVID-19, particularly in high-risk patients.

TGFβ-1 is a potent fibrogenic cytokines capable of inducing organ fibrosis through its promotion of myofibroblast formation [31]. TGFβ-1 is secreted by most immune cells [32]. COVID-19 patients who suffer from the disease for long durations often suffer from severe fibrotic changes in the lungs [33]. These fibrotic pathological findings appear permanent in patients with severe COVID-19. However, prospective studies are important in assessing long-term functional outcomes [33]. Whether the cause of lung fibrosis is due to an aberrant cytokine cascade, viral infection, or both is still a subject of inquiry. Nevertheless, suppressing TGFβ1 with dexamethasone can potentially reduce the formation of myofibroblasts, leading to reduced chances of long-term detrimental outcomes of lung fibrosis.

Finally, we postulated that there may exist some motif commonality between the inflammatory markers of interest and their receptors. Constraint-based multiple sequence alignment revealed that there was a fingerprint region containing highly conserved residues and moderately conserved residues across several inflammatory markers and their receptors [34]. This fingerprint region was found to be highly conserved across NF-κB, INFAR, IL-6R, INF-γ, IFNGR, IL-21R, IL-21, IL-12, TGFβ-1, and CCL5, and is a likely determinant of high affinity binding of dexamethasone (Figure 3). Interestingly, NF-κB is a well-established target for dexamethasone. Yamazaki (2005) demonstrated that dexamethasone inhibited the activity of NF-κB using concentrations between 10 μM and 100 μM [35].

### 3.2. Dexamethasone as a Potential SARS-CoV-2 Inhibitor

We further evaluated dexamethasone as a repurposed therapeutic agent against SARS-CoV-2 infections. Our findings demonstrated that dexamethasone exhibited significant binding affinity for two critical SARS-CoV-2 proteins, 3CLpro and NSP macro X (Figure 2 and Figure 3). Although weaker binding was observed between dexamethasone and the 3CLpro active site, it still provided an avenue to explore analogs of dexamethasone and other established corticosteroids with similar pharmacophores [36,37,38,39].

The macro X domain is a rather novel protein encoded by nonstructural protein 3 (Nsp3) of SARS-CoV-2 [38,40]. The role of Nsp3 in viral replication is still debatable; however, it is believed to play a critical role in ADP–ribose binding, rendering it a promising drug target [40].

3CLpro, another attractive SARS-CoV-2 target [38,39,41], is a key player in viral replication, as it functions by cleaving overlapping polyproteins to mature functional proteins [41]. Suppressing 3CLpro with dexamethasone can halt the synthesis of viral particles and consequently reduce the symptoms and severity of COVID-19.

### 3.3. MD Simulations of Dexamethasone–Cytokine/Chemokine Complex

Next, we employed MD simulations to evaluate the stability of the dexamethasone-cytokine/chemokine complexes. Protein–ligand interaction patterns during MD simulations is essential to understanding the mechanism of inhibition [42]. Furthermore, RMSDs of Cα-atoms of a protein with respect to simulation time give insights into the stability of a drug–protein complex [42].

Our findings demonstrated that IL-1, IL-6, IL-8, IL-12, IL-21, INFα2, TGF TGFβ-1, INF-γ, CXCL8, 3CLpro, and NSP macro X all formed stable complexes with dexamethasone over a simulation time of 50 ns. IL-33 and CCL5 were the only two targets with the least favorable interaction dynamics. RMSDs fluctuating within the 2.0 Å were generally acceptable, which implied the formation of stable protein–inhibitor complexes (Figure 4 and Appendix A) [39,42].

## 4. Conclusions

Our work implied that dexamethasone has the potential to neutralize inflammatory markers, further supporting its life-saving potential in patients with severe manifestations of COVID-19 (Figure 5). Dexamethasone has been known to suppress cytokine and chemokine production, primarily at the transcriptional level, but its role as a direct inhibitor of inflammatory cytokines, chemokines, and their downstream receptors has not yet been discussed in the literature. Therefore, additional studies are necessary to support these findings. Despite the beneficial effects, long-term systemic use of dexamethasone is associated with multiple severe adverse events, and further dose optimization for the treatment of patients with severe manifestations of the COVID-19-induced cytokine storm is an urgent matter of discernment.

## Figures and Tables

**Figure 1 antibiotics-10-01507-f001:**
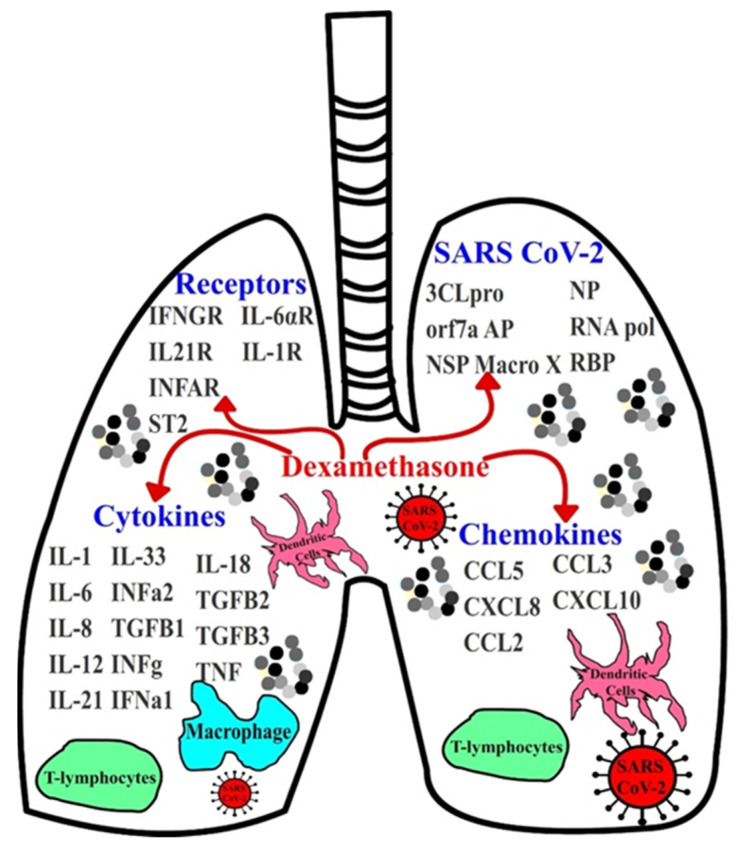
Screening strategy for dexamethasone against several proinflammatory cytokines, chemokines, their receptors, and critical SARS-CoV-2 proteins.

**Figure 2 antibiotics-10-01507-f002:**
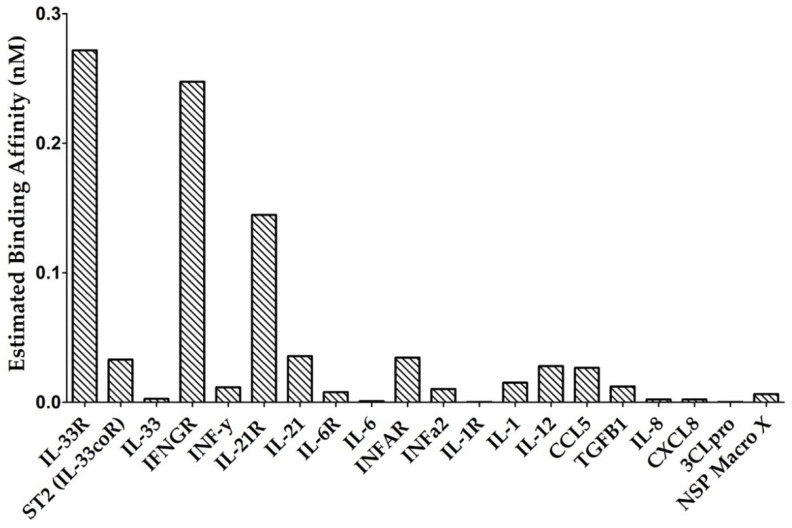
Binding affinities of the virtual screening for dexamethasone against the most promising cytokines, chemokines, receptors, and critical SARS-CoV-2 proteins.

**Figure 3 antibiotics-10-01507-f003:**
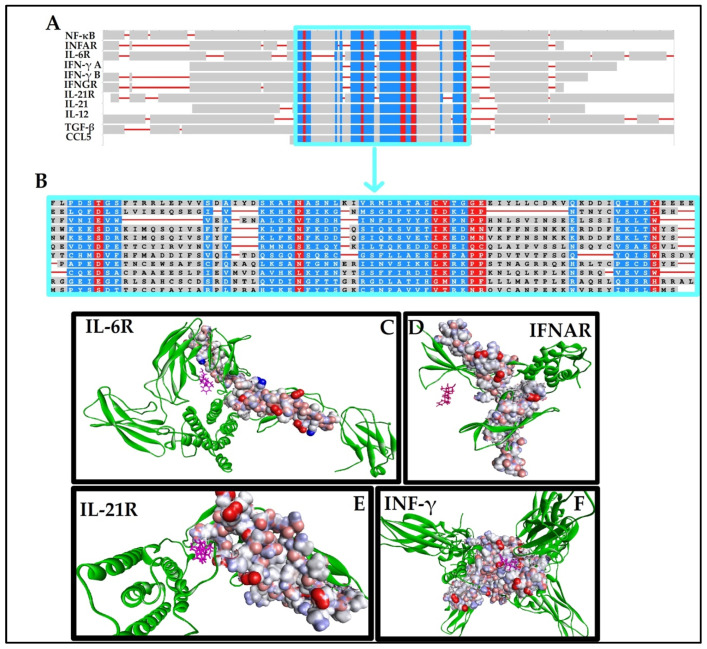
(**A**) The constraint-based multiple alignment (COBALT) for the targets that showed the greatest affinity for dexamethasone. (**B**) A unique region containing highly conserved residues (red) and moderately conserved residues (blue) were identified. These regions were conserved across NF-κB, INFAR, IL-6R, INF-γ, IFNGR, IL-21R, IL-21, IL-12, TGFβ-1, and CCL5. This fingerprint region (red, blue, grey) appeared to be involved in the high affinity biding of dexamethasone, as seen in the 3D ligand interaction diagrams: (**C**) IL-6R; (**D**) IFNAR; (**E**) IL-21R; (**F**) INFG.

**Figure 4 antibiotics-10-01507-f004:**
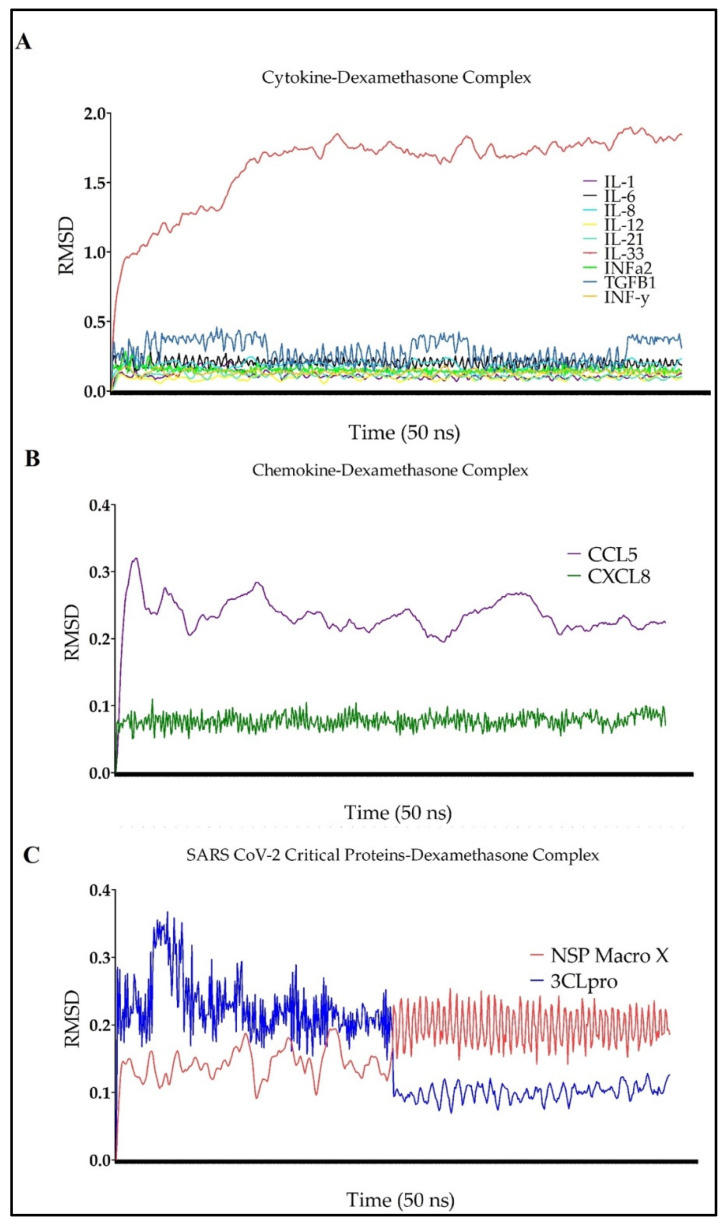
The RMSD of dexamethasone interacting with several promising targets implicated in the cytokine storm: including cytokines (**A**) INF-γ, TGFβ1, INFa2, IL-33, IL-21, IL-12, IL-8, IL-6, and IL-1; (**B**) chemokines CXCL8 and CCL5; and (**C**) SARS-CoV-2 proteins NSP macro X and 3CLpro. RMSD provides insights into the stability of a drug–protein complex.

**Figure 5 antibiotics-10-01507-f005:**
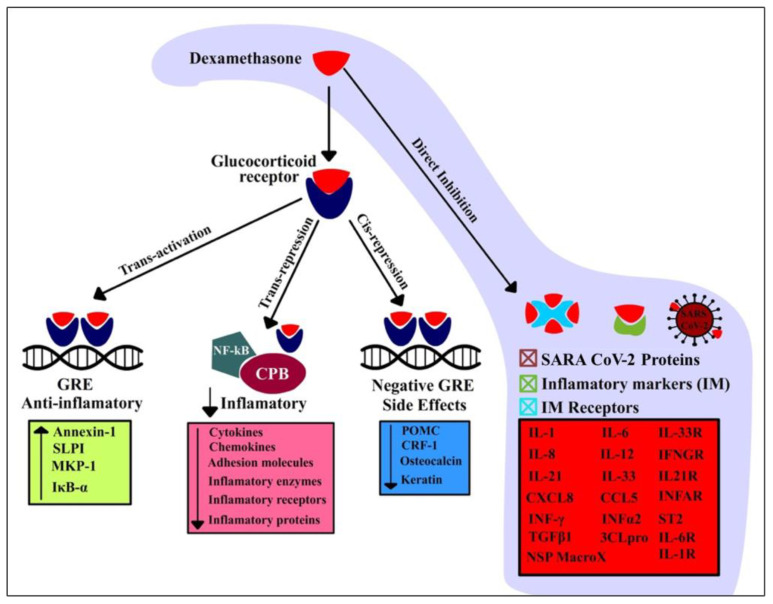
Proposed mechanism of dexamethasone in the suppression of inflammation-mediated lung injury induced through direct binding and inhibition of key inflammatory markers, their downstream receptors, and SARS-CoV-2 proteins. SLPI: human secretory leukocyte protease inhibitor; MAPK-1: mitogen-activated protein kinase; IκBα: nuclear factor of kappa light polypeptide gene enhancer in B-cells inhibitor, alpha; POMC: pro-opiomelanocortin corticotropin releasing factor type 1 (CRF(1)).

**Table 1 antibiotics-10-01507-t001:** The results showed estimated binding energies (ΔG (kcal/mol))**,** estimated inhibitory constant (Ki), and estimated half maximal inhibitory concentration (IC_50_) of dexamethasone for the inflammatory markers, their receptors, and critical SARS-CoV-2 targets.

Category	Target	ΔG [kcal/mol]	Ki [M]	IC_50_ [M]
Receptor	IL-33 receptor (IL-1RAcP)	−11.095	7.400 × 10^−9^	1.472 × 10^−8^
Receptor	IFNG receptor (IFNGR)	−11.040	8.100 × 10^−9^	1.616 × 10^−8^
Receptor	IL-21 receptor (IL21R)	−10.722	1.400 × 10^−8^	2.762 × 10^−8^
Cytokine	IL-21	−8.935	2.820 × 10^−7^	5.640 × 10^−7^
Receptor	INFA2 receptor (INFAR)	−8.917	2.900 × 10^−7^	5.813 × 10^−7^
Receptor	IL-33 receptor (ST2)	−8.890	3.000 × 10^−7^	6.080 × 10^−7^
Cytokine	IL-12	−8.793	3.590 × 10^−7^	7.170 × 10^−7^
Chemokine	CCL5	−8.514	5.740 × 10^−7^	1.150 × 10^−6^
Cytokine	IL-1	−8.429	6.620 × 10^−7^	1.320 × 10^−6^
	TGFβ-1	−8.304	8.180 × 10^−7^	1.640 × 10^−6^
Cytokine	INF-γ	−8.272	8.630 × 10^−7^	1.730 × 10^−6^
Cytokine	INFα2	−8.205	9.670 × 10^−7^	1.930 × 10^−6^
Receptor	IL-6αR-gp130	−8.043	1.300 × 10^−6^	2.543 × 10^−6^
	NSP macro X	−7.929	1.540 × 10^−6^	3.083 × 10^−6^
Cytokine	IL-33	−7.419	3.640 × 10^−6^	7.280 × 10^−6^
Cytokine	IL-8	−7.318	4.320 × 10^−6^	8.640 × 10^−6^
Chemokine	CXCL8	−7.318	4.320 × 10^−6^	8.640 × 10^−6^
Cytokine	IL-6	−6.820	1.000 × 10^−5^	2.000 × 10^−5^
	3CLpro	−5.762	5.970 × 10^−5^	1.190 × 10^−4^
Receptor	IL-1RA receptor (IL−1R)	−5.597	7.900 × 10^−5^	1.570 × 10^−4^
Chemokine	CCL2	NA		
Chemokine	CCL3	N/A		
Chemokine	CXCL10	N/A		
Cytokine	INFα-1	N/A		
	NP	N/A		
Cytokine	TGFβ-2	N/A		
Cytokine	TGFβ-3	N/A		
Cytokine	TNF-α	N/A		
	RNA Pol	N/A		
	RBP	N/A		
	Orf7a AP	N/A		

N/A is representative of binding pockets that had no interaction with dexamethasone, where dexamethasone either had poor shape complementarity with the topology of the pocket or failed to form any favorable intramolecular forces of attraction.

## Data Availability

All the data sources are available in the published manuscript.

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
