# Peer review of "A Closer Look at Dexamethasone and the SARS-CoV-2-Induced Cytokine Storm: In Silico Insights of the First Life-Saving COVID-19 Drug"

_antibiotics, 2021, doi:10.3390/antibiotics10121507_

Round 1

Reviewer 1 Report

The purpose of this manuscript was to explore SARS-Cov-2 induced cytokine storm via docking simulation and MD simulations. However, several major points should be addressed and then revised completely.   Major points:
  • How could the authors carry out docking simulation without the prior position of the binding pocket? How can we know which poses are more reliable (if they apply blind docking simulation)?
  • The authors performed 50-ns MD simulations on the protein-ligand complexes. But why didn't they apply or test longer simulation time? Where is the evidence that 50 ns is enough for the MD simulation conditions?
  • Figure 1: Is it reasonable that all of the reactions occur only in the lung?
  Minor points:
  • All abbreviations should be confirmed. Abbreviations should be spelled out in full at first, and after the first appearance, the abbreviation should be used, such as ICU, RMSD and CPB.
  • Prism 5 --> GraphPad Prism 5 software (GraphPad Software Inc., La Jolla, California, USA)
  • Table 1: All of the data should have the same number of significant figure.
  • Figure 3: The fonts of the legend should be identical.
  • Figure 4: In the Direct Inhibition, two symbols of SARS-CoV-2 proteins and inflammatory markers are identical.
  • All supplementary figures need to be grouped into ONE figure with small alphabets.
  • All citations format should be checked, such as ".[ref]"

Author Response

Reviewer 1

Dear reviewer, your comments were highly constructive and helped to improve the quality of the manuscript. We have now revised the manuscript based on each of your listed concerns and recommendations. Our responses are listed in blue. We look forward to the positive reception and assessment of the revisions.

Major

Q1

How could the authors carry out docking simulation without the prior position of the binding pocket?

Ans

BioSolveIT – SeeSAR suite employs a highly cited algorithm called DoGSiteScorer which is designed specifically for automated detection of protein pockets for small molecule drugs and ligands. DoGSiteScorer is used for druggability assessment of proteins based on its comprehensive ADME properties.

Referenced articles below:

1.         Andrea Volkamer, Daniel Kuhn, Friedrich Rippmann, Matthias Rarey, DoGSiteScorer: a web server for automatic binding site prediction, analysis and druggability assessment, Bioinformatics, Volume 28, Issue 15, 1 August 2012, Pages 2074–2075, https://doi.org/10.1093/bioinformatics/bts310

Q2

How can we know which poses are more reliable (if they apply blind docking simulation)?

Ans

The FlexX docking functionality in SeeSAR uses an incremental construction algorithm to place and fit a ligand into the binding pocket. The ligand is parsed into fragments and a combination of these fragments are positioned into several places in the pocket. The ligand fragments are scored using a pre-scoring algorithm.  The ligand is built up fragment by fragment inside the pocket and the interim solutions are scored against each other. Of the ‘n’ number of possible poses, only the highest ranked scores are returned. Furthermore, poses are ranked based on their overall Hydrogen bond and DEsolvation (HYDE) score. HYDE is an integrated algorithm that addresses the prediction of ligand binding affinity in SeeSAR. HYDE calculates realistic free energies based on two critical parameters: i. desolation of ligand, and ii. physical interactions of ligand in the binding pocket. The novelty of HYDE lies in its ability to intrinsically balance the weighting of hydrogen bonds and desolvation parameters without the need to be trained to specific targets like most other currently described force fields. Finally, HYDE’s ability to visualize the ΔG contributions of the individual atom of a ligand inside the binding pocket allows for reliable discernment and stratification of the most promising poses.

Referenced articles below:

1.     Rarey, M.; Kramer, B.; Lengauer, T.; Klebe, G. A Fast Flexible Docking Method Using an Incremental Construction Algorithm. J. Mol. Biol. 1996, 261 (3), 470–48. https://doi.org/10.1006/jmbi.1996.0477

2.     Warren, G. L.; Andrews, C. W.; Capelli, A. M.; Clarke, B.; LaLonde, J.; Lambert, M. H.; Lindvall, M.; Nevins, N.; Semus, S. F.; Senger, S.; Tedesco, G.; Wall, I. D.; Woolven, J. M.; Peishoff, C. E.; Head, M. S. A Critical Assessment of Docking Programs and Scoring Functions. J. Med. Chem. 2006, 49 (20), 5912–5931. https://doi.org/10.1021/jm050362n

3.     Gastreich, M.; Lilienthal, M.; Briem, H.; Claussen, H. Ultrafast de Novo Docking Combining Pharmacophores and Combinatorics. J. Comput. Aided. Mol. Des. 2006, 20 (12), 717–734.Kubinyi, H.. Success Stories of Computer-Aided Design. Computer Applications in Pharmaceutical Research and Development, 2006, 377-424. https://doi.org/10.1007/s10822-006-9091-x.

4.     Kubinyi, H.. Success Stories of Computer-Aided Design. Computer Applications in Pharmaceutical Research and Development, 2006, 377-424. https://doi.org/10.1002/0470037237.ch16

5.     Reulecke, I.; Lange, G.; Albrecht, J.; Klein, R.; Rarey, M. Towards an Integrated Description of Hydrogen Bonding and Dehydration: Decreasing False Positives in Virtual Screening with the HYDE Scoring Function. ChemMedChem 2008, 3 (6), 885–897. https://doi.org/10.1002/cmdc.200700319

6.     Schneider, N.; Lange, G.; Hindle, S.; Klein, R.; Rarey, M. A Consistent Description of HYdrogen Bond and DEhydration Energies in Protein-Ligand Complexes: Methods behind the HYDE Scoring Function. J. Comput. Aided. Mol.Des. 2013, 27 (1), 15–29. https://doi.org/10.1007/s10822-012-9626-2

Q3

The authors performed 50-ns MD simulations on the protein-ligand complexes. But why didn't they apply or test longer simulation time? Where is the evidence that 50 ns is enough for the MD simulation conditions?

Ans

Protein dynamics can be evaluated and modeled at several length and time scales. Various experimental and computational methods suggest that nanosecond timescales are sufficient to evaluate protein side chain torsions as well as side chain interactions with small molecule compounds like dexamethasone. Nanosecond time scales are therefore appropriate to study the active site dynamics in drug-protein interactions.  Lengthier time scales which extend into the microsecond and millisecond range are more relevant for evaluating protein domain motions and global protein folding and unfolding dynamics.

Referenced articles below:

1.     Srivastava A, Nagai T, Srivastava A, Miyashita O, Tama F. Role of Computational Methods in Going beyond X-ray Crystallography to Explore Protein Structure and Dynamics. Int J Mol Sci. 2018 Oct 30;19(11):3401. doi: 10.3390/ijms19113401. PMID: 30380757; PMCID: PMC6274748.

2.     Perilla JR, Goh BC, Cassidy CK, Liu B, Bernardi RC, Rudack T, Yu H, Wu Z, Schulten K. Molecular dynamics simulations of large macromolecular complexes. Curr Opin Struct Biol. 2015 Apr;31:64-74. doi: 10.1016/j.sbi.2015.03.007. Epub 2015 Apr 4. PMID: 25845770; PMCID: PMC4476923.

Figure 1: Is it reasonable that all of the reactions occur only in the lung?

Ans

Figure 1 is a schematic diagram that summarizes our screening strategy for dexamethasone. Yes, it is plausible that all of these reactions are localized in the lungs. These pro-inflammatory markers are abundant in the lung epithelium and vasculature. SARS-CoV-2 entry receptor is largely expressed in lung epithelium. The pharmacodynamics of dexamethasone involves targets and receptors also abundant in the lungs. Figure 1 is therefore a very realistic representation of the complex interplay taking place in the lung epithelium following infection by SARS CoV-2, and subsequent treatment with dexamethasone.

Minor

Q4

All abbreviations should be confirmed. Abbreviations should be spelled out in full at first, and after the first appearance, the abbreviation should be used, such as ICU, RMSD and CPB.

Prism 5 --> GraphPad Prism 5 software (GraphPad Software Inc., La Jolla, California, USA)

Ans

All abbreviations were standardized and corrected. Reference to GraphPad Prism 5 software 5 was also corrected. Please see revised manuscript.

Q5

Table 1: All of the data should have the same number of significant figure.

All data throughout the manuscript now reflects the same number of significant figures. Please see revised manuscript.

Q6

Figure 3: The fonts of the legend should be identical.

Ans

The fonts and legends were improved and made identical. Please see revised manuscript.

Q7

Figure 4: In the Direct Inhibition, two symbols of SARS-CoV-2 proteins and inflammatory markers are identical.

Ans

Figure 4 has been improved and the issue with the symbols have been corrected fonts and legends were improved and made identical.

Q8

All supplementary figures need to be grouped into ONE figure with small alphabets.

Ans

All supplementary figures were grouped into one figure with small alphabets. Please see revised manuscript.

Q9

All citations format should be checked, such as ".[ref]"

Ans

All citations were cross-checked and updated where necessary. Please see revised manuscript.

Reviewer 2 Report

Authors of the article titled "A Closer Look at Dexamethasone and the SARS CoV-2 Induced 2 Cytokine Storm: In silico Insights of the First Life-Saving COVID-19 3 Drug" reported the study investigating the afinity of dexamathasone to various protein involved in cytokine storm responsible for ARDS developemnt in Sars-Cov2 infected patients. Reported study was well designed. Methodes were adeqatelly chosen. Obtained results have an iportamnt impact on the knowledge about SarsCov2 induced disease. A recommed the article for publication.

Author Response

Reviewer 2

Authors of the article titled "A Closer Look at Dexamethasone and the SARS CoV-2 Induced 2 Cytokine Storm: In silico Insights of the First Life-Saving COVID-19 3 Drug" reported the study investigating the affinity of dexamethasone to various protein involved in cytokine storm responsible for ARDS development in Sars-Cov2 infected patients. Reported study was well designed. Methods were adequately chosen. Obtained results have an important impact on the knowledge about SarsCov2 induced disease. A recommend the article for publication.

Dear reviewer, your comments were highly constructive and significantly helped to improve the quality of the revised manuscript. We have now put forth a much more comprehensive manuscript based on the consolidated comments and recommendations from all reviewers. Below is a list of the improvements that were made.

1.)   Table 1 now reflects the quantitative values for all proteins used in the study.

2.)   Figure 2 reflects the quantitative values for all proteins used in the study.

3.)   Figure 3 was added and shows promising results from a multiple sequence alignment identifying a conserved region implicated in dexamethasone binding

4.)   Figure 5 color code was improved

5.)   The discussion section was improved with reference to several published works that further support the findings of our study.

6.)   The supplementary figures were merged into a single figure.

7.)   The references were updated to reflect the revisions in the manuscript

8.)   The grammatical errors, fonts, SI units, and significant figures were also corrected.

Please find attached the revised manuscript.

I look forward to your reception and assessment of the revised manuscript.

Reviewer 3 Report

Morgan et al., in the manuscript entitled “A Closer Look at Dexamethasone and the SARS CoV-2 Induced Cytokine Storm: In silico Insights of the First Life-Saving COVID-19 Drug” has predicted the possible inhibitory role of Dexamethasone on SARS-CoV2 replication through binding to several cytokines and chemokines and viral proteins. This role is in addition to the already known transcription regulatory role of Dexamethasone, which could qualify this drug as a very potent agent against SARS CoV2 infection. The manuscript is written well overall; however I have few queries and suggestions to improve it further.

Major comments

  1. Authors say in the page 4, that Figure 1 is about the docking of the structurally binding of this drug and the proteins. However, I see only a cartoon of lungs with protein names written. As IL33R, IFNGR and IL21R are showing strong binding, docking of these proteins-drug is worth showing in main figures, so that the manuscript becomes interesting to readers.
  2. Have have shown the docking of RNA pol and Dexamethasone in Fig. S13 but calculation of the other parameters for this interaction is missing from the table1 or figures 2, 3. Please take care of this.
  3. Do all the cytokines shown to bind with Dexamethasone and the viral proteins share sequence or domain similarity as all of them are binding to Dexamethasone? It would be a nice idea to include a parallel analysis of the sequence or domain similarity among the candidate proteins that could probably binding to Dexamethasone.
  4. Authors have suggested that suppressing Mpro with Dexamethasone can halt the synthesis of viral particles. However, same is not supported by the data given in the table 1 and Figure 2. Even the figure 3 does not strongly support that Mpro and Dexamethasone could make a strong protein-drug complex. How do authors justify this statement in that case?
  5. Through MD simulations, authors showed that IL33 and CCL5 were the only two targets with least favorable interaction dynamics. However, in Figure 2 CCL5 has much better estimated binding affinity (nM) for Dexamethasone than many other cytokines, chemokines and SARS CoV2 proteins. Additionally, Table 1 shows that IL33 and CCL5 also have lower binding energy (hence better) than many cytokines and chemokines. Could authors explain this variation?

Minor comments

  1. Take care of minor typos and grammatical errors across the manuscript. Such as in page 5, line 211-215.
  2. Discrepancy in abbreviations. Switching between IFN and INF for the interferon
  3. Please use scientific terms for the symbol alpha, beta and gamma.

Author Response

Reviewer 3

Dear reviewer, your comments were highly constructive and helped to improve the quality of the manuscript. We have now revised the manuscript based on each of your listed concerns and recommendations. Our responses are listed in blue. We look forward to the positive reception and assessment of the revisions.

Major

Q1

Authors say in the page 4, that Figure 1 is about the docking of the structurally binding of this drug and the proteins. However, I see only a cartoon of lungs with protein names written. As IL33R, IFNGR and IL21R are showing strong binding, docking of these proteins-drug is worth showing in main figures, so that the manuscript becomes interesting to readers.

Ans

Figure 3 was added to the manuscript. Figure 3 illustrates the Constraint-Based Multiple Alignment (COBALT) for the targets that showed the greatest affinity for dexamethasone. A unique region containing highly-conserved residues were identified. This region was highly conserved across NF-κB, INFAR, IL-6R, INF-γ, IFNGR, IL-21R, IL-21, IL-12, TGFβ-1 and CCL5. This fingerprint region appears to be involved in the high affinity biding of dexamethasone as depicted in the 3-D ligand interaction diagrams (C-F).

Q2

Have shown the docking of RNA pol and Dexamethasone in Fig. S13 but calculation of the other parameters for this interaction is missing from the table1 or figures 2, 3. Please take care of this.

Ans

This issue was corrected. Table 1. now includes all the quantitative values for all proteins that were screened in the study. In regard to fig 4., not all proteins were candidates for MD simulations with dexamethasone. The exclusion criteria included proteins that were too large (>25 kDa) for computing, and proteins with binding pockets that had no favorable intermolecular interactions with dexamethasone i.e. physically incapable of fitting inside the binding pocket.

Q3

Do all the cytokines shown to bind with Dexamethasone and the viral proteins share sequence or domain similarity as all of them are binding to Dexamethasone? It would be a nice idea to include a parallel analysis of the sequence or domain similarity among the candidate proteins that could probably binding to Dexamethasone.

Ans

Figure 3 is a tremendous addition to the manuscript. Figure 3 shows the Constraint-Based Multiple Alignment (COBALT) for the targets that showed the greatest affinity for dexamethasone. A unique region containing highly conserved residues were identified. This region was highly conserved across NF-κB, INFAR, IL-6R, INF-γ, IFNGR,IL-21R, IL-21, IL-12, TGFβ-1 and CCL5. This fingerprint region appears to be involved in the high affinity biding of dexamethasone as depicted in the 3-D ligand interaction diagrams (C-F).

Q4

Authors have suggested that suppressing Mpro with Dexamethasone can halt the synthesis of viral particles. However, same is not supported by the data given in the table 1 and Figure 2. Even the figure 3 does not strongly support that Mpro and Dexamethasone could make a strong protein-drug complex. How do authors justify this statement in that case?

Ans

Our results referenced in table 1 demonstrate that 3CLpro (Mpro) has an estimated binding energy (ΔG) of -5.762 kcal/mol, which translates to an estimated affinity of 0.169 μM. This is a very relevant and significant free energy of binding. At face value, figure 2. may appear “misleading” as the estimated affinities were expressed in units of nM (y-axis). It is important to note that many FDA approved drugs and bioactive natural products display affinities in the μM range. However, with the advent of sophisticated computer-aided drug design pipelines, more and more FDA approved drugs are being optimized with affinities in the nM and even pM range. Furthermore, to support the relevance of our values, Yamazaki (2005) demonstrated that dexamethasone directly inhibited the activity of NF-κB using concentrations between 10 μM and a 100 μM.

References:

1.     Yamazaki T, Tukiyama T, Tokiwa T. Effect of dexamethasone on binding activity of transcription factors nuclear factor-kappaB and activator protein-1 in SW982 human synovial sarcoma cells. In Vitro Cell Dev Biol Anim. 2005 Mar-Apr;41(3-4):80-2. 

Q5

Through MD simulations, authors showed that IL33 and CCL5 were the only two targets with least favorable interaction dynamics. However, in Figure 2 CCL5 has much better estimated binding affinity (nM) for Dexamethasone than many other cytokines, chemokines and SARS CoV2 proteins. Additionally, Table 1 shows that IL33 and CCL5 also have lower binding energy (hence better) than many cytokines and chemokines. Could authors explain this variation?

Ans

In regard to MD simulations, RMSDs (Root Mean Square Deviations) of Cα-atoms of a protein with respect to simulation time give insights into the stability of a drug-protein complex.  It is possible that a drug may have relatively high binding affinity but unfavorable interaction dynamics due to a multitude of factors driven by the global flexibility and stability of the protein target in its native environment. RMSDs fluctuating within the 2.0 Å are generally acceptable, which implies the formation of stable protein-inhibitor complexes. As such, the ideal effective drug must have sufficiently high affinity and favorable interaction dynamics. MD simulation tend to reveal more information about interaction dynamics of the drug-protein complex in its native solvent environment.

1.)   AlAjmi MF, Azhar A, Owais M, Rashid S, Hasan S, Hussain A, et al. Antiviral potential of some novel structural analogs of standard drugs repurposed for the treatment of COVID-19. Journal of biomolecular structure & dynamics. 2020:1-13. Epub 2020/07/31. doi: 10.1080/07391102.2020.1799865.

Minor

Q6

Take care of minor typos and grammatical errors across the manuscript. Such as in page 5, line 211-215.

Ans

The typos and grammatical errors throughout the manuscript were corrected. Please see revised manuscript.

Q7

Discrepancy in abbreviations. Switching between IFN and INF for the interferon

Ans

Abbreviations were corrected throughout the manuscript. Please see revised manuscript.

Q8

Please use scientific terms for the symbol alpha, beta and gamma.

Ans

Scientific terms for symbol alpha, beta and gamma were corrected throughout the manuscript. Please see revised manuscript.

Reviewer 4 Report

This work “A Closer Look at Dexamethasone and the SARS CoV-2 Induced Cytokine Storm: In silico Insights of the First Life-Saving COVID-19 Drug” (antibiotics-1462208) presented by Chih-Wen Shu et al. provides an interesting perspective for the study of SARS COV-2: the possible interactions between Dexamethasone and critical molecules in a series of cell storms (cytokines, their receptors, and SARS-CoV-2 proteins) from the perspective of molecular simulation. Before being accepted, some concerns should be addressed.

MAJOR POINTS:

  1. The entire 3.1 section can be moved to the discussion section.
  2. The docking results between Dexamethasone and other proteins should be presented.
  3. Where is the “discussion” part?
  4. Considering that this work is an in silico study, the authors should dig more detailed info from other experimental results to support their findings. This part could be presented in the “discussion” part.

MINOR POINTS:

line 135. “Data analysis and correlation plots were generated using, Prism 5. ”---Typos, correct it.

Author Response

Reviewer 4

Major

Q1

The entire 3.1 section can be moved to the discussion section.

Section 3.1 is in now listed in the results and discussion section. The discussion was significantly improved.

Q2

The docking results between Dexamethasone and other proteins should be presented.

Table 1. was revised to reflect the results from the docking of dexamethasone and the other proteins presented in the study. CCL2, CCL3, CXCL10, INFα-1, NP, TGFβ-2,TGFβ-3, TNF-α, RNA Pol, and RBP had suboptimal binding pockets and showed no interaction (NaN) with dexamethasone.  

Q3

Where is the “discussion” part?

The Results and Discussion section are combined and can be found from lines 152 to 408.

Q4

Considering that this work is an in silico study, the authors should dig more detailed info from other experimental results to support their findings. This part could be presented in the “discussion” part.

The scope of this manuscript discusses our hypothesis which attempts to correlate the role of dexamethasone as a therapeutic with the potential to directly bind and suppress several key cytokines, chemokines and functionally critical SARS-CoV-2 proteins. Uncontrolled over-production of cytokines and chemokines is the primary cause of Acute Respiratory Distress Syndrome (ARDS) observed in patients with severe COVID-19 i.e. cytokine storm. Although the focus of our work is primarily computational there are several published experimental studies that are in direct agreement with our hypothesis. We have now integrated these experimental studies in our discussion.

Yamazaki et al. (2005) demonstrated that dexamethasone inhibited NF-κB (nuclear factor kappa-light-chain-enhancer of activated B cells) binding activity at 10 μM as and a 100 μM in SW982 using electrophoretic mobility shift assay (EMSA). In 2008, Mogensen et al. mentioned that that the array of dexamethasone targets with regards to cytokine production are at best, only partially characterized. They further showed using Enzyme-Linked Immunoassay (ELISA) that dexamethasone significantly suppressed IL-6 and IL-8 mRNA production. The strongest experimental evidence yet comes from Monick et al. (1994) ELISA based study which demonstrated that dexamethasone inhibited IL-1 and TNF release of IL-6 and IL-8 from cells in a dose-dependent manner. They suggested that in this instance dexamethasone inhibition is not by disrupting the signal transduction pathway responsible for increasing or decreasing expression of IL-1 or TNF receptors. But instead they postulate that “it is possible that dexamethasone as decreases cytokine production by simply freezing the receptor ligand on the cell surface.”  Kadowaki et al. (2019) demonstrated that pH-stimulated CXCL8 production in Human airway smooth muscle cells (ASMCs) was inhibited by dexamethasone in a dose-dependent manner. Finally, Zhang et al. (2021) Western blot study showed that levels of intracellular IL-6 modestly increased when cells were treated with exogenous recombinant IL-6. But treatment with recombinant IL-6 and dexamethasone resulted in a 70% decrease in immunoreactive IL-6 proteins when compared to the untreated cells. Taken together, there is overwhelming evidence in support of our hypothesis supporting the direct inhibitory role of cytokines and chemokines.

References:

1.)   Yamazaki T, Tukiyama T, Tokiwa T. Effect of dexamethasone on binding activity of transcription factors nuclear factor-kappaB and activator protein-1 in SW982 human synovial sarcoma cells. In Vitro Cell Dev Biol Anim. 2005 Mar-Apr;41(3-4):80-2. 

2.)   Zhang Y, Hu S, Wang J, Xue Z, Wang C, Wang N. Dexamethasone inhibits SARS-CoV-2 spike pseudotyped virus viropexis by binding to ACE2. Virology. 2021 Feb;554:83-88.

3.)   Mogensen TH, Berg RS, Paludan SR, Østergaard L. Mechanisms of dexamethasone-mediated inhibition of Toll-like receptor signaling induced by Neisseria meningitidis and Streptococcus pneumoniae. Infect Immun. 2008 Jan;76(1):189-97.

4.)   Monick MM, Aksamit TR, Geist LJ, Hunninghake GW. Dexamethasone inhibits IL-1 and TNF activity in human lung fibroblasts without affecting IL-1 or TNF receptors. Am J Physiol. 1994 Jul;267(1 Pt 1):L33-8. 

Minor

Q4

line 135. “Data analysis and correlation plots were generated using, Prism 5. ”---Typos, correct it.

They typos in line 135 were corrected. Please see revised manuscript.

Round 2

Reviewer 1 Report

The manuscript has been improved considerably but it needs to be done further.

Major points

  • Based on the materials and methods, the authors did just docking simulation and calculated RMSD. Is it possible to investigate the potential mechanism of cytokine storm and dexamethasone?

  • The authors enumerate the latest information of IL-21, IL-1, IL-8, IL-33, CCL5, CXCL8, IFNa-2, TGFb-1, etc. They should link their findings with the information, and then address major issues related with the main purpose of this research.

  • Both Figure 2 and Table 1 show same results of estimated binding affinity. Thus, figure 2 is redundant data. Furthermore, Table 2 should be re-arranged for easy understanding by adding a column having "Cytokine", "Chemokine" and "Receptor".

  • According to the simulation setting including target proteins, ligands and pH condition, the suitable time of MD simulations is different, such as 50 ns, 100 ns and 150 ns. What evidence was 50-ns simulation suitable from?

  • The authors should describe the results separately, pro-inflammatory-response-related part and critical viral proteins part.

  • In conclusion section, the authors just did docking simulation of dexamethasone with some cytokines and chemokines. But how did they propose the entire mechanism of the drug?

  • "2.3. Cyotkines, Chemokines..." section should be moved to the first place because materials should be described in general without specific reasons.

Minor points

  • Sentences should not begin with numbers (lines 85, 101)

  • Which one is correct? INF-g (line 24, 120, 238, 260, 270, 277, Table 1) vs IFN-g (lines 51, 157)

  • In "2.3. Cyotkines, Chemokines..." section, which database were the PDB files retrieved from? When was it accessed?

  • Line 113, RMSD should be described in full name.

  • In Table 1, 1) should "NaN" be used? It indicates "Not a Number".  So, "NA" is commonly used. 2) "1.30E-06" and "4.32E-06" should be corrected with the same number of significant figure. 3) "1.570 E-04" should be corrected to "1.570E-04".

  • Why were many words italicized? (Lines 119, 122, 126, 239, 241, 246, 273,

  • In line 241, "between10" -> "between 10"

  • Which one is correct? nsp3 or NSP3, TGFb-1 vs TGFb1

  • In figure 3B, what does "fingerprint" mean? If necessary, please visualize it in figure 3A. In figure 3C-F, please describe what 3D models in green and surface structures mean, respectively.

  • in figure 4C, Prototeins -> Proteins

  • In figure 4 legend, (A) and (B) should not be in bold.

  • In line 295, the title of Table S1 is not same with the one of the Table S1.

  • In line 273, Figures S1-S20 -> Figure S1a-t

  • In Figure 5 legend, "induced through of" should be correctly revised. What are those words?, GRE, SLP1, MKP-1 and IkB-a.

Author Response

Reviewer 1 – Round 2

Dear reviewer, your comments were highly constructive and helped to improve the quality of the manuscript. We have now revised the manuscript based on each of your listed concerns and recommendations. Our responses are listed in blue. We look forward to the positive reception and assessment of the revisions.

Major

Q1

 Based on the materials and methods, the authors did just docking simulation and calculated RMSD. Is it possible to investigate the potential mechanism of cytokine storm and dexamethasone?

Ans

The focus of our work was not designed with the intention to unravel a comprehensive mechanism  between the cytokine storm and dexamethasone. Instead, we hypothesize that multiple cytokines and chemokines implicated in the COVID-19 induced cytokine storm are key targets for dexamethasone –- which has not yet been described in the literature.

Mogensen et al. (2008) stated that the array of dexamethasone targets with regards to cytokine production are at best, only partially characterized. Furthermore, the comprehensive anti-inflammatory mechanism by which dexamethasone reduces significant risk of mortality in patients with severe manifestations of COVID-19 is still poorly understood (Wagner, 2021). We therefore sought to establish  a direct connection between dexamethasone  and several key inflammatory markers that are implicated in the COVID-19 induced cytokine storm.  Our hypothesis is strongly supported by a multitude of published experimental studies which all allude to the direct binding and inhibition of several inflammatory markers by dexamethasone.

Yamazaki et al. (2005) demonstrated that dexamethasone inhibited NF-κB (nuclear factor kappa-light-chain-enhancer of activated B cells) binding activity at 10 μM and a 100 μM in SW982 using electrophoretic mobility shift assay (EMSA). Interestingly, our Constraint-Based Multiple Alignment (COBALT) revealed a unique region containing highly-conserved residues that appears essential to high affinity binding of dexamethsone. This ‘fingerprint’ region was highly conserved across NF-κB, INFAR, IL-6R, INF-γ, IFNGR, IL-21R, IL-21, IL-12, TGFβ-1 and CCL5.

Mogensen et al. (2008) further showed using Enzyme-Linked Immunoassay (ELISA) that dexamethasone significantly suppressed IL-6 and IL-8 mRNA production. Monick  et al. (1994) ELISA based study found that dexamethasone inhibited IL-1 and TNF release of IL-6 and IL-8 from cells in a dose-dependent manner. They suggested that dexamethasone inhibition is not by disrupting the signal transduction pathway responsible for increasing or decreasing expression of IL-1 or TNF receptors. But instead “it is possible that dexamethasone decreases cytokine production by simply freezing the receptor ligand on the cell surface.” 

Kadowaki et al. (2019) also demonstrated that pH-stimulated CXCL8 production in Human airway smooth muscle cells (ASMCs) was inhibited by dexamethasone in a dose-dependent manner.

Finally, Zhang et al. (2021) Western blot study showed that levels of intracellular IL-6 modestly increased when cells were treated with exogenous recombinant IL-6. But treatment with recombinant IL-6 and dexamethasone resulted in a 70% decrease in immunoreactive IL-6 proteins when compared to the untreated cells.

Taken together, there is overwhelming evidence in support of our hypothesis supporting the direct inhibitory role of cytokines and chemokines. We have re-worded our introduction and conclusion to closer reflect our hypothesis – “Our work implies that dexamethasone has the potential to directly neutralize inflammatory markers further supporting its life saving potential in patients with severe manifestations of COVID-19.”

Please see revised manuscript.

References:

1.)   Yamazaki T, Tukiyama T, Tokiwa T. Effect of dexamethasone on binding activity of transcription factors nuclear factor-kappaB and activator protein-1 in SW982 human synovial sarcoma cells. In Vitro Cell Dev Biol Anim. 2005 Mar-Apr;41(3-4):80-2. 

2.)   Zhang Y, Hu S, Wang J, Xue Z, Wang C, Wang N. Dexamethasone inhibits SARS-CoV-2 spike pseudotyped virus viropexis by binding to ACE2. Virology. 2021 Feb;554:83-88.

3.)   Mogensen TH, Berg RS, Paludan SR, Østergaard L. Mechanisms of dexamethasone-mediated inhibition of Toll-like receptor signaling induced by Neisseria meningitidis and Streptococcus pneumoniae. Infect Immun. 2008 Jan;76(1):189-97.

4.)   Monick MM, Aksamit TR, Geist LJ, Hunninghake GW. Dexamethasone inhibits IL-1 and TNF activity in human lung fibroblasts without affecting IL-1 or TNF receptors. Am J Physiol. 1994 Jul;267(1 Pt 1):L33-8. 

5.)   Wagner C, Griesel M, Mikolajewska A, Mueller A, Nothacker M, Kley K, Metzendorf MI, Fischer AL, Kopp M, Stegemann M, Skoetz N, Fichtner F. Systemic corticosteroids for the treatment of COVID-19. Cochrane Database Syst Rev. 2021 Aug 16;8(8):CD014963. doi: 10.1002/14651858.CD014963.

Q2

The authors enumerate the latest information of IL-21, IL-1, IL-8, IL-33, CCL5, CXCL8, IFNa-2, TGFb-1, etc. They should link their findings with the information, and then address major issues related with the main purpose of this research.

Ans

In the discussion section of the manuscript we reiterate the significance of  our findings with respect to the most current literature on inflammatory markers and their role in the cytokine storm. We also discuss potential challenges that may arise from targeting these inflammatory markers with dexamethasone. Please see revised manuscript.

Q3

Both Figure 2 and Table 1 show same results of estimated binding affinity. Thus, figure 2 is redundant data. Furthermore, Table 2 should be re-arranged for easy understanding by adding a column having "Cytokine", "Chemokine" and "Receptor".

Ans

A  new column highlighting "Cytokine", "Chemokine" and "Receptor” categories was added to the manuscript.  We believe that Figure 2 is a critical part of the manuscript and is not totally  redundant to Table 1. This is because ‘binding affinity’ is not totally superimposable with ‘binding energies’. The energy released during bond formation between dexamethasone and the binding pocket is calculated as the binding energy. Whereas, the efficiency by which dexamethasone interacts with the binding pocket is termed as affinity. Lower binding energies suggest higher affinities. Nevertheless, to reduce the overlap of content between Table 1 and Figure 2 we omitted the targets from Figure 1 that showed no interaction with dexamethasone. Please see revised manuscript. Please see revised manuscript.

Q4

According to the simulation setting including target proteins, ligands and pH condition, the suitable time of MD simulations is different, such as 50 ns, 100 ns and 150 ns. What evidence was 50-ns simulation suitable from?

Ans

Protein dynamics can be evaluated and modeled at several length and time scales. Various experimental and computational methods suggest that nanosecond timescales are sufficient to evaluate protein side chain torsions as well as side chain interactions with small molecule compounds like dexamethasone. Nanosecond time scales are therefore appropriate to study the active site dynamics in drug-protein interactions. 

Lengthier  time scales which extend into the microsecond and millisecond range are more relevant for evaluating protein domain motions and global protein folding and unfolding dynamics. Please see revised manuscript.

Referenced articles below:

1.     Srivastava A, Nagai T, Srivastava A, Miyashita O, Tama F. Role of Computational Methods in Going beyond X-ray Crystallography to Explore Protein Structure and Dynamics. Int J Mol Sci. 2018 Oct 30;19(11):3401. doi: 10.3390/ijms19113401. PMID: 30380757; PMCID: PMC6274748.

2.     Perilla JR, Goh BC, Cassidy CK, Liu B, Bernardi RC, Rudack T, Yu H, Wu Z, Schulten K. Molecular dynamics simulations of large macromolecular complexes. Curr Opin Struct Biol. 2015 Apr;31:64-74. doi: 10.1016/j.sbi.2015.03.007. Epub 2015 Apr 4. PMID: 25845770; PMCID: PMC4476923.

Q5

The authors should describe the results separately, pro-inflammatory-response-related part and critical viral proteins part.

Ans

The results are described separately. Section 3.1 discusses pro-inflammatory-response and its relation to dexamethasone. 3.2 discusses critical viral proteins and their relation to dexamethasone. Please see revised manuscript.

Q6

In conclusion section, the authors just did docking simulation of dexamethasone with some cytokines and chemokines. But how did they propose the entire mechanism of the drug?

Ans

We have re-worded our introduction and conclusion to closer reflect our hypothesis – “Our work implies that dexamethasone has the potential to directly neutralize inflammatory markers further supporting its life saving potential in patients with severe manifestations of COVID-19.” Please see revised manuscript.

Q7

2.3. Cytokines, Chemokines..." section should be moved to the first place because materials should be described in general without specific reasons.

Ans

2.3  was moved up in the materials and methods section. 2.3 is now 2.1. Please see revised manuscript.

Minor

Q8

Sentences should not begin with numbers (lines 85, 101)

Ans

All sentences beginning with numbers were corrected. Please see revised manuscript.

Q9

Which one is correct? INF-g (line 24, 120, 238, 260, 270, 277, Table 1) vs IFN-g (lines 51, 157)

Ans

INF-γ is the widely accepted abbreviation for the cytokine while INFGR is conventionally used to denote the receptor. The inconsistencies have been corrected. Please see revised manuscript.

Q10

In "2.3. Cyotkines, Chemokines..." section, which database were the PDB files retrieved from? When was it accessed?

Ans

Accession dates from the Protein Data Bank was added to the manuscript for the cytokines, chemokines, receptors, and SARS-CoV-2 proteins. Please see revised manuscript.

Q11

Line 113, RMSD should be described in full name.

Ans

RMSD was described in full at first instance. Please see revised manuscript.

Q12

In Table 1, 1) should "NaN" be used? It indicates "Not a Number".  So, "NA" is commonly used. 2) "1.30E-06" and "4.32E-06" should be corrected with the same number of significant figure. 3) "1.570 E-04" should be corrected to "1.570E-04".

Ans

We replaced NaN with NA.

1.30E-06" and "4.32E-06" were corrected with the same number of significant figures.

1.570 E-04" was corrected to "1.570E-04".

Please see revised manuscript.

Q13

Why were many words italicized? (Lines 119, 122, 126, 239, 241, 246, 273,. In line 241, "between10" -> "between 10"

Ans

This text formatting issue was corrected. Please see revised manuscript.

Q14

Which one is correct? nsp3 or NSP3, TGFb-1 vs TGFb1

Ans

Nsp3 is the correct acronym. This was corrected. Please see revised manuscript.

Q15

In figure 3B, what does "fingerprint" mean? If necessary, please visualize it in figure 3A. In figure 3C-F, please describe what 3D models in green and surface structures mean, respectively.

Ans

In the manuscript we describe the  fingerprint region as the highly conserved region across the proteins with the highest affinities for dexamethasone. The highlighted surface structures (red, blue, grey) shown in Figure 3(C-F), depict the fingerprint region which appears to be critical to high affinity binding of dexamethasone. Please see revised manuscript.

Q16

in figure 4C, Prototeins -> Proteins

Ans

Figure 4C was corrected Prototeins -> Proteins

Q17

In figure 4 legend, (A) and (B) should not be in bold.

Ans

Figure 4 legend A and B are no longer bold.

Q18

In line 295, the title of Table S1 is not same with the one of the Table S1.

Reviewer 4 Report

The authors handled the referee's concerns in a satisfactory way.

Thanks. 

Author Response

Round 2 – Revised Manuscript

Dear reviewer, your assessment of the manuscript was extremely constructive and helped to improve the overall quality in this most recently revised manuscript. The revisions were based on comprehensive feedback from all the reviewers. Below you find a list of changes that were made to improve the overall quality of the manuscript

1.)   Table 1 now reflects the quantitative values for all proteins used in the study. A column was also added to more readily identify the category of each target i.e. cytokine, chemokine, receptor, SARS-CoV-2 protein.

2.)   Figure 2 reflects the quantitative values for all proteins used in the study. All values are of the same significant figures

3.)   Figure 3 was added and shows promising results from a multiple sequence alignment identifying a conserved region implicated in dexamethasone binding.

4.)   Figure 5 color code  and legend was improved

5.)   The discussion section was further improved with reference to several experimental works that further support the findings of our hypothesis.

6.)   The supplementary figures were merged into a single figure.

7.)   The references were updated to reflect the revisions in the manuscript

8.)   The grammatical errors, fonts, SI units, and significant figures were also corrected.

9.)   Figure 4 was improved

10.) The discussion section was improved for better organization and clarity.

11.) The abstract and conclusion sections were improved to better reflect the central theme of our hypothesis.

12.) The methods section was reorganized and  improved for better readability

Please find attached the revised manuscript.

I look forward to your reception and assessment of the revised manuscript.
